ecology/theoretical biology/computational biology

growth, von Bertalanffy, Gompertz, mixed-effects models

**Author for correspondence:**
Simone Vincenzi
e-mail: simon.vincenz@gmail.com

# Biological and statistical interpretation of size-at-age, mixed-effects models of growth

## Simone Vincenzi[1], Dusan Jesensek[2] and Alain J. Crivelli[3]

[1]Independent Scholar
[2]Tolmin Angling Association, Most Na Soci, Tolmin, Slovenia
[3]Station Biologique de la Tour du Valat, Le Sambuc 13200, Arles, France

 SV, 0000-0002-8436-8608

The differences in life-history traits and processes between organisms living in the same or different populations contribute to their ecological and evolutionary dynamics. We developed mixed-effect model formulations of the popular size-at-age von Bertalanffy and Gompertz growth functions to estimate individual and group variation in body growth, using as a model system four freshwater fish populations, where tagged individuals were sampled for more than 10 years. We used the software Template Model Builder to estimate the parameters of the mixed-effect growth models. Tests on data that were not used to estimate model parameters showed good predictions of individual growth trajectories using the mixed-effects models and starting from one single observation of body size early in life; the best models had $R^2 > 0.80$ over more than 500 predictions. Estimates of asymptotic size from the Gompertz and von Bertalanffy models were not significantly correlated, but their predictions of size-at-age of individuals were strongly correlated ($r > 0.99$), which suggests that choosing between the best models of the two growth functions would have negligible effects on the predictions of size-at-age of individuals. Model results pointed to size ranks that are largely maintained throughout the lifetime of individuals in all populations.

## 1. Introduction

Understanding the causes of within- and among-population variation in vital rates of organisms, such as their probability of survival, growth, migration, and reproduction, life histories (i.e. how vital rates vary together and the trade-offs among them), and population dynamics (i.e. how the number of individuals in a population changes over time) is a central topic in ecology and evolutionary biology.

Among vital rates, body growth is probably the one that has historically received the most attention, since survival, sexual maturity, reproductive success, and movement and migration are frequently associated with, or affected by, growth and body size [1]. Therefore, variation in growth within and among populations contributes to their ecological and evolutionary dynamics [2–4], and a better understanding of growth will always be an important problem in biology.

Mathematical and statistical models have been a staple of research in ecology and evolutionary biology for many decades; they have been used for, among other goals, statistical inference, identification, quantification, and prediction of biological and environmental processes and their realizations, and decision-making in species management and conservation [5]. When studying the growth of organisms in either applied or theoretical contexts, the identification of a functional form that can reasonably approximate the observed trajectories is a crucial first step in the development and application of a useful growth function. Structured or parametric models for describing and predicting the growth of organisms that continue to increase in size after sexual maturity (i.e. indeterminate growers, [6]) imply a basic functional form: size increases monotonically with time and it usually tends to an upper asymptote later on in life. Since the modelling of mechanisms is more informative than curve fitting (e.g. it allows to predict outside of the range of data observed and test hypotheses on the determinants of growth), ideally the parameters of the growth function should describe some of the physiological or life-history processes that affect growth. However, depending on their assumptions and parametrizations, growth functions fitted on the same data can provide different—sometimes in disagreement—insights into the growth process [7].

For vertebrates, the most widely used growth functions for size-at-age have been the Gompertz (GGF) [8,9] and the von Bertalanffy growth functions (vBGF) [10], two special cases of the Richards growth function [11]. The other macro-group of growth functions is auto-regressive functions, which use the individual's previous size to predict its size at the next time point (see [12] for a modern study of those functions). Either the GGF or the vBGF are reasonable choices when modelling the size-at-age of organism with initial slow growth, which then increases in speed before levelling off toward adult value. Their most popular parametrizations include three parameters, one of which—the asymptotic size—is common between the GGF and the vBGF.

In the vast majority of applications of growth models in ecology and evolutionary biology, the parameters of growth functions have been estimated at the population level, and interpreted as those of an average individual in the population [13]. However, estimates of parameters at the population level neither describe nor explain the large variation in growth often observed in organisms living in the same population. There are many examples of individuals (e.g. fish) living in the same population and of similar age with very different—sometimes hugely different—sizes [14,15]; for most investigations, we are thus interested in each individual, and not population- or group-average, growth trajectory. Understanding which biological and environmental processes determine group and individual variation in growth may also improve predictions of future growth and size of individuals and populations, which are valuable for the conservation and management of species [16,17].

However, understanding the nature and contribution of sources of variation in growth faces a number of experimental, methodological and computational challenges. First, we need longitudinal data (i.e. multiple observations of the same individual through time) to tease apart growth variation emerging from persistent differences among individuals or among groups from variation due to stochastic processes [12,18], and to predict individual and (average) group growth trajectories. Especially for long-lived and elusive organisms such as fish, the collection of longitudinal data can take many years and much effort (e.g. individuals are usually tagged with a unique ID identifier and are then re-captured over the next months or years), when at all possible. As marine scientist John Sheperd said: 'Managing fisheries is hard: it's like managing a forest, in which the trees are invisible and keep moving around' (http://jgshepherd.com/thoughts/).

Longitudinal data are thus often sparse, and data for a particular individual are unlikely to be sufficient for the estimation of parameters of its individual-specific growth model. However, we may leverage data of other individuals that are thought to be similar to support the estimation of model parameters for the data-poor individual (i.e. the concept of 'borrowing strength', [19]). Models in which all members in a group influence the estimate of each effect are called either hierarchical, random-effects, multi-level or mixed-effects models [20].

Random effects in mixed-effects models are realizations of a stochastic process. The assumption of common statistical distribution implies a dependence between random effects, which means that the estimate of the random effect for an individual is influenced by data and estimates of random effects for all other individuals relating to the same factor or group (say, for an organism, year-of-birth, sex, location,

or, at a coarser grain, the whole population when modelling the growth of individuals from different populations). Since in mixed-effects models those realizations will be pulled toward the mean of the group (shrinkage, [19]), the realizations that are strongly supported by data contribute more information to the statistical distribution of the effects, and we are then less likely to over-interpret processes that are the result of small sample sizes. Modelling and estimating random effects also address the lack of independence between repeated measurements of the same individuals [20]. Mixed-effects models of growth thus provide an intuitive framework for estimating heterogeneity of growth among individuals or groups of individuals, approximate their growth trajectories, and predict their future sizes [21].

Second, growth models of organisms are often—if not always—nonlinear, and the estimation of parameters of nonlinear models is intrinsically more computationally demanding than for linear models. In practice, however, in order to investigate multiple parametrizations of complex models, we need fast and reliable approaches to parameter estimation [13]. Although parameter estimation in mixed-effects models can still require considerable work, the many—and in some cases groundbreaking—advances in theory, algorithms and software of the last few years have made the development of mixed-effect models and the estimation of their parameters much easier (i.e. the optimization algorithms more reliably converge) and faster than it used to be [22,23].

Here, we take advantage of those recent theoretical and computational advances to propose mixed-effect formulations of the von Bertalanffy and Gompertz growth functions that can help model individual and group variation in the growth of organisms. Since [24] developed von Bertalanffy equations for individual growth, several growth models describing or accounting for individual and group variation in growth, along with algorithms for their fitting, have been developed [25]. For instance, [26] recently developed a Bayesian state-space framework, there applied to the vBGF, for modelling growth that allows for intrinsic individual variation in traits, a shared environment, process stochasticity and measurement error. Contreras-Reyes *et al.* [27] developed a framework that comprised modelling of individual variability of size-at-age, Bayesian inference and distribution of errors from the Student-*t* model.

Our work on growth is motivated by the long-term study of populations of marble trout (*Salmo marmoratus*), brown trout (*Salmo trutta* L.), and rainbow trout (*Oncorhynchus mykiss*) living in Slovenian streams. Marble, brown and rainbow trout populations were sampled each June and September for more than 10 years (2004–2015), and the data show substantial variation in size-at-age of individuals [28–30]. Vincenzi *et al.* [30,31] showed that fast-growth allows trout populations to recover after massive mortality events, such as those caused by flash floods, since fecundity tends to increase with fish size. In addition, since growth and body size are heritable [32], there is potential for the evolution of growth rates toward faster growth in populations affected by massive mortality events [33,34]. Therefore, models of growth that can predict growth trajectories for individuals, groups (e.g. organisms born before or after extreme events) and populations will greatly benefit our understanding of the resilience of populations to extreme events and of the evolution of body growth in the affected populations.

In this paper, we: (i) propose mixed-effects, individual-based formulations of the well-known and widely used von Bertalanffy and Gompertz growth functions, and show how their parameters can be efficiently estimated with Template Model Builder, an open-source statistical software package for fitting nonlinear, mixed-effects statistical models using the Laplace approximation [35]; (ii) use four salmonid populations living in Slovenian streams as a model system to test the ability of the mixed-effects growth models (i.e. von Bertalanffy and Gompertz growth functions with predictors of their parameters, including random effects) to predict unobserved size-at-age of individuals; (iii) test whether the life-history and biological processes determining the observed individual variation in growth (in particular, the maintenance of size hierarchies among individuals throughout their lifetime), are inferred differently when using the von Bertalanffy or the Gompertz growth function.

# 2. Material and methods

## 2.1. Study area and species description

Our study system includes the marble trout (*Salmo marmoratus*) populations of Lower Idrijca [LIdri_MT] and Upper Idrijca [UIdri_MT] [28], rainbow trout (*Oncorhynchus mykiss*) population of Lower Idrijca [LIdri_RT] [36], and brown trout (*Salmo trutta* L.) population of Upper Volaja [UVol_BT] [29]. All streams are located in western Slovenia [36]. The streams Lower Idrijca and Upper Idrijca are a few hundred metres apart and are separated by a dam. In LIdri, marble trout [LIdri_MT] live in sympatry with rainbow trout [LIdri_RT] [36,37]. Both UIdri_MT and UVol_BT live in allopatry, that is no other

fish species live in the streams. The LIdri_RT population was created in the 1960s [36] and the UVol_BT population in the 1920s [29] by stocking rainbow and brown trout, respectively. Both populations have been self-sustaining since their inception. Further details on the demographic and life-history traits of these salmonid populations, along with their conservation status and current management practices, can be found in [36].

## 2.2. Sampling

Populations were sampled bi-annually in June and September. The first sampling for LIdri_MT, LIdri_RT and UIdri_MT was in June 2004, and in September 2004 for UVol_BT. If captured fish had length $L >$ 115 mm, and had not been previously tagged or had lost a previously applied tag, they received a Carlin tag [38], and their age was determined by reading scales [39]. Genetic tagging was then used to identify fish that had received a new tag after losing theirs between sampling occasions [36]. Fish are 0+ (juveniles) in the first calendar year of life, 1+ in the second year and so on. Sub-yearling marble, rainbow and brown trout are smaller than 115 mm in June and September, so fish were tagged when at least aged 1+. Sampling protocols are described in greater details in [28] and [29]. Total number of fish aged 1+ or older sampled from 2004 to 2015 was 1371 for LIdri_MT, 670 for UIdri_MT, 250 for LIdri_RT and 2636 for UVol_BT.

## 2.3. Growth functions

Here, we introduce two of the most widely used growth functions in ecology, evolutionary biology and fishery science: the von Bertalanffy and the Gompertz growth functions. Although there have been calls for moving beyond the von Bertalanffy and Gompertz growth functions for statistical (e.g. negative correlation between parameter estimates) and life-history (e.g. they do not account for the change in energy allocation after sexual maturity) reasons [40], they are still popular—and almost the default choices—among modellers.

### 2.3.1. The von Bertalanffy growth function

The vBGF has been used to model the growth of fish [41], mammals [42], snakes [43], birds [44] and many other species and taxa. von Bertalanffy hypothesized that the growth of an organism results from a dynamic balance between anabolic and catabolic processes [10]. If $W(t)$ denotes mass at time $t$, the von Bertalanffy assumption is that anabolic factors are proportional to surface area, which scales as $W(t)^{2/3}$, and that catabolic factors are proportional to mass. If $a$ and $b$ denote these scaling parameters, then the rate of change of mass is

$$\frac{dW}{dt} = aW(t)^{2/3} - bW(t). \tag{2.1}$$

If we assume that mass and length (i.e. size), $L(t)$, are related by $W(t) = \rho L(t)^3$ with $\rho$ corresponding to density, then calculus shows that

$$\frac{dL}{dt} = q - kL, \tag{2.2}$$

where $q = a/3\rho$ and $k = b/3\rho$.

Setting $L_\infty = q/k$ to be the asymptotic size and $L(0) = L_0$ to be the initial size, two forms of the solution are

$$L(t) = L_\infty(1 - e^{-kt}) + L_0 e^{-kt} \tag{2.3}$$

and

$$L(t) = L_\infty(1 - e^{-k(t-t_0)}), \tag{2.4}$$

where $t_0$ is the hypothetical age at which length is equal to 0.

The forms of the vBGF in equations (2.3) and (2.4), which are derived following the assumption of anabolism scaling with mass to the 2/3 power and mass scaling with the cube of size, are referred to as the 'specialized' vBGF [45]. For simplicity, in this paper, we will simply use the term vBGF for the 'specialized' vBGF.

If $L(t) > L_\infty$, the rate of growth $k$ (in time$^{-1}$ units) is negative, so asymptotic size is the upper limit of size, which is only attained in the limit of infinite time. In this work, we will use the formulation of the vBGF of equation (2.4), which has three parameters: $L_\infty$, $k$ and $t_0$. Although the definition of asymptotic size in the vBGF introduces an explicit linear relationship on the log scale between $k$ and $L_\infty$ (i.e. $\log(L_\infty) = \log(q) - \log(k)$, according to equation (2.4)), in this work, we do not treat $L_\infty$ as equal to $q/k$ (see [7] for a formulation of the function explicitly including the link between $L_\infty$, $q$ and $k$), but we let the correlation between $L_\infty$ and $k$ at the whole population and at the individual level emerge from data.

### 2.3.2. The Gompertz growth function

The Gompertz growth function (GGF) [8,9] has been used to model the growth of a variety of species and taxa, from plants to birds and fish growth, to tumour and bacterial growth [46,47]. Contrary to the vBGF, which was developed by von Bertalanffy from physiological principles, the Gompertz curve was first proposed by Benjamin Gompertz [9] for modelling mortality curves, and later adopted for studies of growth when it was empirically found that the GGF could well describe the growth trajectories of many species [8]. Therefore, the biological interpretation of the parameters of the GGF is harder and tends to be more *a posteriori* than that of the von Bertalanffy function (but see [48] for an example of biological interpretation of the Gompertz model for tumour growth), and we introduce the parameters of the GGF mostly as curve-fitting parameters.

The GGF has been used in a variety of parametrizations, with some formulations that have parameters that are more interpretable than in others [46]. One commonly found parametrization for the GGF is

$$L(t) = A \exp(-\exp(-k_G(t - T_i)), \tag{2.5}$$

where $L(t)$ is size at time $t$; $A$ (we might also call this parameter $L_\infty$ like in the vBGF, since the two parameters both represent the asymptotic size); $k_G$ (in time$^{-1}$ units) is a coefficient of growth affecting the slope; and $T_i$ is time at inflection, which in this formulation occurs when 37% of the final growth has been reached. $T_i$ shifts the growth curve horizontally without changing its shape and therefore is a location parameter, while $A$ and $k_G$ are shape parameters.

## 2.4. Parameter estimation and individual variation

In this section, we will mostly make reference to the vBGF for describing mixed-effects models of growth and the estimation of their parameters, but all assumptions and methods are also applicable to the GGF and other similar size-at-age growth functions.

In the vast majority of applications of the vBGF and of the GGF, $L_\infty$, $k$ and $t_0$ (and $A$, $k_G$ and $T_i$) have been estimated at the population level (i.e. without accounting for individual heterogeneity in growth) starting from cross-sectional data, and interpreted as the growth parameters of an average individual in the population (i.e. $L_\infty$ is the asymptotic size of an average or modal individual of the population or species). In fishery science, vBGF's $k$ and $L_\infty$ and estimates of adult mortality are commonly used descriptors of the life-history strategies of fish populations [49]. Age-structured stock assessment methods can also be based on size-at-age estimations, which are often derived from parameters of the von Bertalanffy growth model for that species [50].

Vincenzi *et al.* [7,13] presented formulations of the vBGF specific for longitudinal data where $L_\infty$, $k$ and $t_0$ may be allowed to be a function of shared predictors and individual random effects. Since $k$ and $L_\infty$ must be non-negative, it is convenient to use a log-link function. We thus set for individual $i$ in group $j$

$$\begin{cases} \log(k_{ij}) = \alpha_0 + \alpha_1^{(j)} + \alpha_2 x_{ij} + \sigma_u u_{ij} \\ \log(L_\infty^{(ij)}) = \beta_0 + \beta_1^{(j)} + \beta_2 \delta_{ij} + \sigma_v v_{ij}, \\ t_0^{(ij)} = \gamma_0 + \gamma_1^{(j)} + \gamma_2 \phi_{ij} + \sigma_z \zeta_{ij}, \end{cases} \tag{2.6}$$

where $u \sim N(0,1)$, $v \sim N(0,1)$ and $z \sim N(0,1)$ are the standardized individual random effects, $\sigma_u$, $\sigma_v$ and $\sigma_z$ are the standard deviations of the statistical distributions of the random effects, $\alpha_1^{(j)}$, $\beta_1^{(j)}$ and $\gamma_1^{(j)}$ are group (i.e. categorical) effects (e.g. species, population, sex, year-of-birth), and the other parameters are defined as in equation (2.4). The continuous predictors $x_{ij}$, $\delta_{ij}$ and $\phi_{ij}$ in equation (2.6) (e.g. population density, temperature, proxies of food quality and availability) do not need to enter linearly into the predictor, and the terms $\alpha_2 x_{ij}$, $\beta_2 \delta_{ij}$ and $\gamma_2 \phi_{ij}$ may be replaced by a more general function

$f(x_{ij}; \tau)$, where $\tau$ denotes a set of parameters to be estimated. For the GGF, the parameters $A$, $k_G$ and $T_i$ are set the same way as in equation (2.5).

Length of individual $i$ in group $j$ at age $t$ is for the von Bertalanffy function

$$L_{ij}(t) = L_\infty^{(ij)}(1 - e^{-k_{ij}(t - t_0^{(ij)})}) + \varepsilon, \tag{2.7}$$

and for the Gompertz function

$$L_{ij}(t) = A^{(ij)} \exp(-\exp(-k_G^{(ij)}(t - T_i^{(ij)}) + \varepsilon, \tag{2.8}$$

where $\varepsilon$ is the observation error, which is time-invariant and normally distributed with mean 0 and variance $\sigma_\varepsilon^2$.

According to equations (2.7) and (2.8), a positive correlation between $L_\infty^{(ij)}$ and $k_{ij}$ (from now on we will refer to them as $L_\infty$ and $k$ at the individual level) and $A^{(ij)}$ and $k_G^{(ij)}$ ($A$ and $k_G$ at the individual level) indicates that size ranks tend to be maintained through the life of individuals, while a negative correlation indicates that size ranks tend not to be maintained [7,13].

## 2.5. Model fitting

It has often been difficult, if not impossible using standard approaches, to estimate parameters for many of the proposed growth models using data on individual growth trajectories in natural settings (i.e. in the wild). In addition to noisy and sparse data, and even in the presence of a large amount of data, a highly parametrized nonlinear model may be only weakly statistically identifiable. Several modelling tools to fit hierarchical models are now available, such as platform-independent BUGS [51], JAGS [52], Stan [53] and, among many others, the *nlme*, *lme4*, *NIMBLE*, *brms* packages and associated functions in R [22,54–56] or *PyMC3* in Python [57].

However, when dealing with a large number of random effects, nonlinear models, and missing or noisy data, some of those methods may fail to converge or take a very long time to converge, thus limiting the number of explorations that can be carried out. Since many models are typically fitted when investigating biological process, it is always convenient, and often necessary in practice, to use algorithms and software that allow for rapid model exploration, for instance by reducing the time needed for the optimization algorithm to converge or by finding a compromise between obtaining the full posterior distribution of parameters (more computationally expensive) or only its summary statistics (less computationally expensive).

A tool that has been recently developed for fitting highly parametrized mixed-effects models, and which we used for fitting models in the present study, is Template Model Builder (TMB, [35]). TMB is a general random-effects tool integrated in R that was inspired by ADMB (Automatic Differentiation Model Builder [58]), an open-source statistical software package for fitting highly parametrized nonlinear statistical models with or without random effects. TMB can fit generic random-effects models using an empirical Bayes approach that evaluates and maximizes the Laplace approximation of the marginal likelihood [59] using automatic differentiation. TMB computes standard errors of parameter estimates and of predictions using the delta method [60], offers easy access to parallel computations, and is very flexible in model formulation. Recent developments allow the estimation of parameters of Bayesian models using TMB [61].

## 2.6. Statistical analyses

Prediction of future observables has long been included as an aspect of statistics, but it has been much less prominent than parametric statistical inference [62]. More recently, it has been proposed that the proper use of statistical models is the prediction of future observations, and not uncertainty around estimates of model parameters [63]. In this work, we will evaluate models from the point of view of both inference (parameter estimates and their biological interpretations) and predictive performance.

Since exploratory investigations often prevent the use of null-hypothesis testing, and multiple comparisons increase the 'researcher degrees of freedom', including the choice of convenient hypotheses to test [64], apart from some specific tests of correlation, we present and discuss our model results from a qualitative point of view, that is without formal null-hypothesis testing.

We found empirically that for both the GGF and vBGF, the fitting algorithm converges more reliably when size data are evenly spaced in time. For the analyses of growth, we thus used only September data (unique fish sampled in September: LIdri_MT, $n = 784$; UIdri_MT, $n = 502$, LIdri_RT, $n = 109$, UVol_BT,

$n = 2434$). Vincenzi *et al.* [28] found no or minor effects of population density, water temperature, or sex on growth in Slovenian populations of marble, rainbow, and brown trout. In this study, we pooled all data from different populations together and use only group variables as candidate factors for explaining variation in growth. We introduced group predictors as fixed effects (since treating a factor with just a few levels as 'random' may generate imprecise estimates of the associated standard deviation [65]) to test whether they improved model performance with respect to a model with no predictors other than random effects. In particular, we included as predictors (i) *Species* (three-level predictor: marble trout, *MT*; brown trout, *BT*; rainbow trout, *RT*), and (ii) *Population* (four-level predictor: *LIdri_MT*, *UIdri_MT*, *LIdri_RT* and *UVol_BT*) as a group (i.e. categorical) variable. We tested for vBGF and GGF all the combination of *Species*, *Population* and *Constant* (i.e. no predictors) for the three parameters of either growth function. We also experimented with *Cohort* (i.e. year-of-birth) as a group predictor and with two-way interactions between *Species*, *Population* and *Cohort*.

We tested the predictive ability of the candidate vBGF and GGF models as follows. For each model, we first tested for convergence of the TMB algorithm and computed the Akaike information criterion (AIC [66]) when fitting the model on the whole dataset. We then: (i) identified fish that were sampled more than three times; (ii) randomly sampled one-third of them (test dataset, 201 unique individuals); (iii) deleted from the dataset all observations except the first one from each individual fish in the validation sample; (iv) if the TMB algorithm converged, estimated the parameters of the vBGF and GGF for each individual including those in the validation sample; (v) predicted the observations in the test dataset ($n$ between 522 and 541 from the 201 unique individuals); and (vi) estimated or computed accuracy measures, such as $R^2$ with respect to the $1:1$ prediction-observation line, and maximum error for the test sample. We repeated (ii)–(vi) five times and then computed the mean of the accuracy measures for each vBGF and GGF model.

Following [7,13], we estimated the correlation between $L_\infty$ and $k$, and $A$ and $k_G$, to get insights into potential processes driving the observed variation in growth. We also estimated the correlation between $L_\infty$ and $A$ at the individual level within populations when using the same models for the two growth functions, since the values of $L_\infty$ and $A$ are both estimates of asymptotic size.

Results are fully reproducible. Data and R code are at https://github.com/simonevincenzi/Growth_Models.

# 3. Results

There were 60 individuals sampled three or more times in LIdri_MT, 7 in LIdri_RT, 90 in UIdri_MT and 513 in UVol_BT. Empirical growth trajectories showed substantial individual variation in growth rates and size-at-age (figure 1).

Most models including either *Cohort* or interacting predictors never converged or converged only for some of the five replicates, and we thus dropped them from the set of candidate models. A number of different vBGF and GGF models had basically the same predictive accuracy, and their AICs computed on the full dataset were fairly close as well (table 1). For 12 out of the 24 models, $R^2$ with respect to the $1:1$ predicted-observed line was greater than 0.80 (table 1). The overall correlation (Pearson's $r$) between $L_\infty$ and $A$ when using the same model (i.e. same predictors for the 'equivalent' parameters) and pooling together all population-specific estimates was 0.51 ($p < 0.01$) (figure 2). However, within-population correlations between $L_\infty$ and $A$ were all non-significant.

The GGF model with *Population* as predictor for all three parameters was the model with the best AIC computed on the whole dataset ($\varDelta$AIC with the second-best model = 39.08) (table 1). The equivalent-in-predictors vBGF model showed estimates of asymptotic size that were similar only for UVol_BT to those provided by the GGF (figure 3). In both vBGF and GGF models with *Population* as predictor for all three parameters, $L_\infty$ and $k$, and $A$ and $k_G$ at the individual level were positively correlated within populations (figure 4).

GGF and vBGF predictions of sizes in the test datasets were remarkably similar to each other, with Pearson's correlation between GGF and vBGF predictions for the same models (i.e. same predictors used in either growth function) on average greater than 0.99 (both prediction and parameter estimate uncertainties are provided in the online resources associated with this paper).

However, despite the strong positive correlation between GGF and vBGF model predictions, for the model with *Species* predicting $A$ and $L_\infty$ for GGF and vBGF, and *Population* predicting the other two parameters for either function (i.e. the best model in terms of average predictive accuracy on test datasets for both the GGF and vBGF, table 1), the estimated $A$ and $L_\infty$ were largely different from

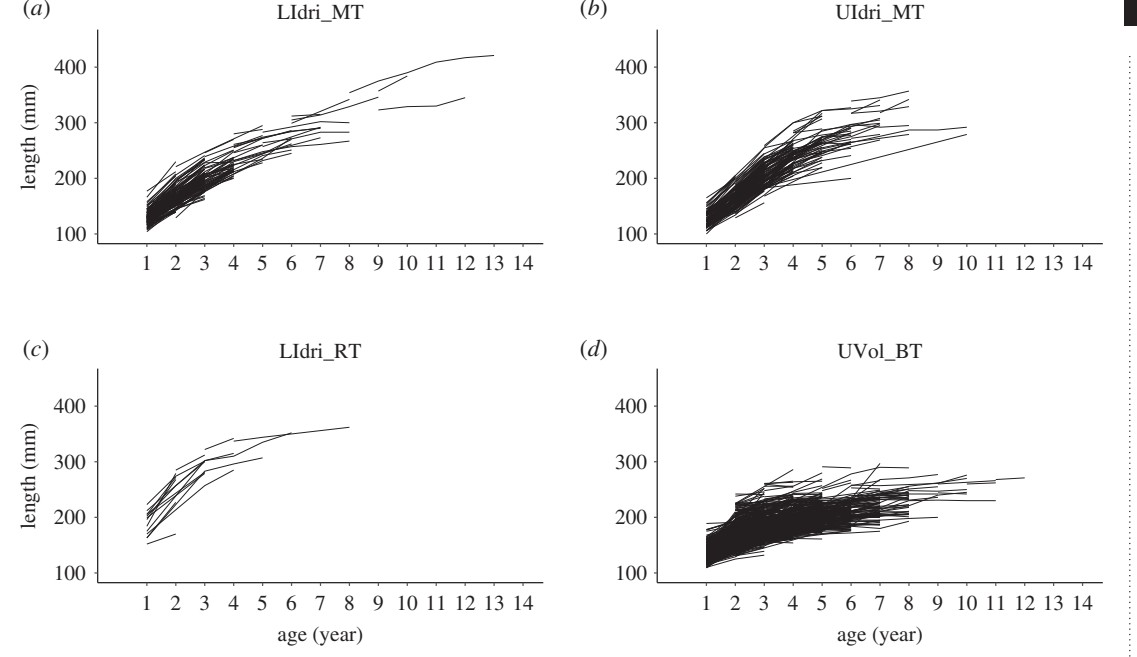

**Figure 1.** Growth trajectories of fish sampled more than once between 2004 and 2015 (only September samplings) in the populations of LIdri_MT ($n = 210$ unique fish), UIdri_MT ($n = 209$), LIdri_RT ($n = 17$) and UVol_BT ($n = 1323$).

each other for the population of LIdri_RT (GGF:$\hat{A}$ [s.e.] = 721 [48] mm, vBGF: $\hat{L}_\infty$ [s.e.] = 338 [23] mm) and similar for the other three populations (figure 5). In addition, for LIdri_RT, the correlation between $A$ and $k_G$ in the GGF was negative ($r = -0.56$, $p < 0.01$), and between $L_\infty$ and $k$ in the vBGF was positive ($r = 0.91$, $p < 0.01$).

Across models, a few individuals living in UVol_BT and UIdri_MT were consistently those with the largest prediction errors. Those individuals had growth trajectories that were unusual with respect to the common growth trajectories in their populations (figure 6). Both in the many trajectories that were predicted accurately and in the few that were not, the growth trajectories estimated by the GGF and vBGF models with the same predictors were basically indistinguishable (figure 7).

# 4. Discussion

We found that mixed-effects models based on either the von Bertalanffy or Gompertz growth functions were able to largely capture the individual variation in growth among fish living in four distinct freshwater salmonid populations. Among the models we tested, the predictive performances on test datasets of the best von Bertalanffy and Gompertz mixed-effects models were largely equivalent, although their estimates of asymptotic size were often substantially different. Finally, parameter estimates for both growth functions point to strong maintenance of size hierarchies over time in each of the four salmonid populations.

## 4.1. Growth processes

Individuals living in the same environment, and especially those of species with growth after sexual maturity, often vary in their body growth rate and size-at-age. In the four trout populations that have been investigated in this work, the size of the smallest age-1 fish was approximately 50% of the size of the biggest age-1 fish. Growth trajectories in fish are often consistent through time, that is individuals that are small early in life are likely to be among the smallest years later [13,67]. Vincenzi et al. [13] showed that a positive correlation between asymptotic size and growth rate points to the maintenance of size hierarchies through the lifetime of organisms, that is if individual $a$ is bigger than individual $b$ at age $t$, individual $a$ is likely to be bigger than individual $b$ at time $t + 1 \ldots t + n$. For all four populations, we found a positive correlation between asymptotic size and growth rate at the individual level when using either growth function.

**Table 1.** Ranking of 24 models (12 for the von Bertalanffy growth function and 12 for the Gompertz growth function) according to their mean $R^2$ with respect to the 1 : 1 predicted-observed line over five random test datasets. $R^2$ [s.d.] is the standard deviation of the five $R^2$ calculated for each model. In parentheses, the predictors for the growth function parameters, either *Species*, *Population* (*Pop*) or *Constant* (i.e. no predictors, *Const*). Function is either the von Bertalanffy (*vBGF*) or the Gompertz (*GGF*) growth function. The best model when using the whole dataset according to the Akaike information criterion (AIC) was the Gompertz growth function model with $A$ (*Pop*), $k_G$ (*Pop*), $T_i$ (*Pop*).

| model | function | $R^2$ [mean] | $R^2$ [s.d.] | AIC |
|---|---|---|---|---|
| $L_\infty$(*Species*), $k$(*Pop*), $t_0$(*Pop*) | *vBGF* | 0.852 | 0.02 | 54717.75 |
| $L_\infty$(*Pop*), $k$(*Pop*), $t_0$(*Pop*) | *vBGF* | 0.851 | 0.02 | 54717.12 |
| $L_\infty$(*Pop*), $k$(*Pop*), $t_0$(*Const*) | *vBGF* | 0.847 | 0.02 | 54771.37 |
| $A$(*Species*), $k_G$(*Pop*), $T_i$(*Pop*) | *GGF* | 0.846 | 0.03 | 54749.06 |
| $L_\infty$(*Species*), $k$(*Pop*), $t_0$(*Const*) | *vBGF* | 0.846 | 0.02 | 54775.51 |
| $A$(*Pop*), $k_G$(*Pop*), $T_i$(*Pop*) | *GGF* | 0.846 | 0.03 | 54678.04 |
| $A$(*Pop*), $k_G$(*Const*), $T_i$(*Pop*) | *GGF* | 0.845 | 0.02 | 56017.61 |
| $A$(*Species*), $k_G$ (*Const*), $T_i$(*Pop*) | *GGF* | 0.843 | 0.02 | 54768.57 |
| $L_\infty$(*Pop*), $k$(*Const*), $t_0$(*Pop*) | *vBGF* | 0.842 | 0.02 | 54942.05 |
| $L_\infty$(*Species*), $k$(*Const*), $t_0$(*Pop*) | *vBGF* | 0.838 | 0.02 | 54962.76 |
| $L_\infty$(*Const*), $k$(*Pop*), $t_0$(*Pop*) | *vBGF* | 0.810 | 0.02 | 55546.41 |
| $A$(*Const*), $k_G$(*Pop*), $T_i$(*Pop*) | *GGF* | 0.801 | 0.02 | 55647.87 |
| $A$(*Pop*), $k_G$(*Pop*), $T_i$(*Const*) | *GGF* | 0.770 | 0.03 | 56047.32 |
| $A$(*Species*), $k_G$(*Pop*), $T_i$(*Const*) | *GGF* | 0.768 | 0.03 | 56222.17 |
| $L_\infty$(*Pop*), $k$(*Const*), $T_i$(*Const*) | *vBGF* | 0.714 | 0.04 | 56551.44 |
| $A$(*Pop*), $k_G$(*Const*), $T_i$(*Const*) | *GGF* | 0.712 | 0.04 | 56568.42 |
| $L_\infty$(*Species*), $k$(*Const*), $t_0$(*Const*) | *vBGF* | 0.708 | 0.04 | 56577.43 |
| $A$(*Species*), $k_G$(*Const*), $T_i$(*Const*) | *GGF* | 0.707 | 0.04 | 56596.06 |
| $L_\infty$(*Const*), $k$(*Const*), $T_i$(*Pop*) | *vBGF* | 0.700 | 0.06 | 56767.01 |
| $A$(*Const*), $k_G$(*Pop*), $T_i$(*Const*) | *GGF* | 0.700 | 0.05 | 56843.89 |
| $A$(*Const*), $k_G$(*Const*), $T_i$(*Pop*) | *GGF* | 0.698 | 0.06 | 57097.36 |
| $L_\infty$(*Const*), $k$(*Const*), $t_0$(*Const*) | *vBGF* | 0.679 | 0.05 | 57567.55 |
| $A$(*Const*), $k_G$(*Const*), $T_i$(*Const*) | *GGF* | 0.675 | 0.05 | 57570.43 |
| $L_\infty$(*Const*), $k$(*Pop*), $t_0$(*Const*) | *vBGF* | 0.665 | 0.06 | 56896.74 |

In freshwater trout, the primary type of intra-specific competition for resources seems to be interference competition for space [7], probably due to their strong territoriality. In interference competition, bigger individuals reduce the access to resources, such as space and food, of smaller individuals, and may also live longer. High heritability of growth [32], maternal decisions on the timing and location of spawning [68], and dominance established early in life [69] are all processes that in combination or by themselves may explain the maintenance of size ranks throughout fish lifetime.

## 4.2. Prediction of growth trajectories

There is a rich literature on the comparison between growth function for prediction of unobserved data and inference on growth processes in species [70–73]. When developing mathematical and statistical models in biology and ecology, and in particular when those models are used for making predictions of unobserved or future realizations of biological processes, we face trade-offs between model complexity, interpretability of model parameters, ease of parameter estimation, and accuracy of predictions. It is common for more complex models, either in number of predictors, how the predictors enter the model (e.g. nonlinearly) or the algorithm used to estimate model parameters, to provide higher accuracy (here, both the ability of the model to explain observed data and make

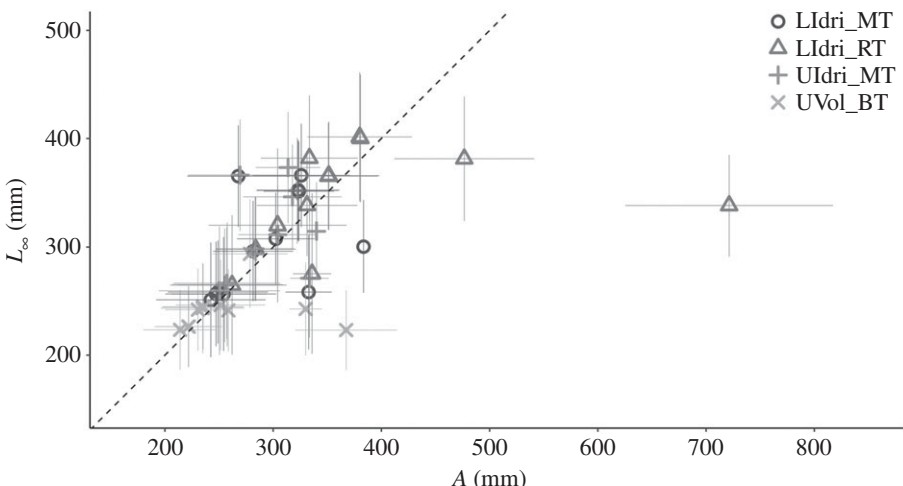

**Figure 2.** Point estimates of $L_\infty$ and $A$ for the same models (i.e. same predictors for the equivalent model parameters) along with their 95% confidence interval when using the whole dataset. The mean [min, max] of the ratio $L_\infty/A$ across the estimates for the 12 von Bertalanffy and 12 Gompertz growth models that converged in all five training/testing splits of the dataset were 1.03 [0.78, 1.36] for LIdri_MT, 0.96 [0.47, 1.14] for LIdri_RT, 1.06 [0.13, 0.81] for UIdri_MT and 0.96 [0.143, 0.60] for UVol_BT. The dashed line is the 1 : 1 line.

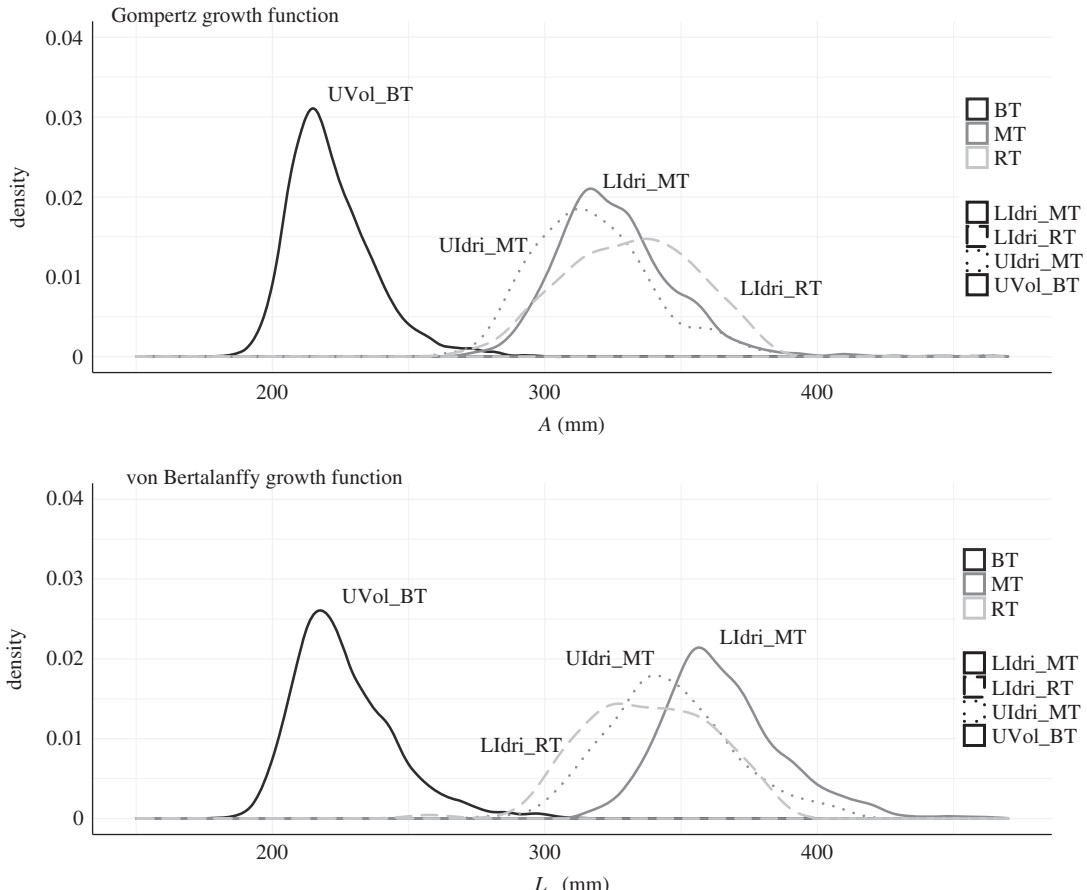

**Figure 3.** Empirical distribution of population-specific individual $L_\infty$ and $A$ for the best GGF (and equivalent von Bertalanffy growth function) model according to AIC (i.e. *Population* predicting all three parameters) when using the whole dataset. GGF (mean (s.d.)): LIdri_MT, 326.00 mm (22.52); UIdri_MT, 317.79 (22.71); LIdri_RT, 331.29 (23.24), UVol_BT, 221.66 (15.61). Von Bertalanffy growth function: LIdri_MT, 365.93 mm (23.82); UIdri_MT, 346.00 (24.11); LIdri_RT, 338.15 (23.22), UVol_BT, 226.44 (18.70).

correct predictions about future observables). However, higher accuracy may come at the expense of ease of parameter estimation, interpretability of model predictors and parameters (i.e. to what degree the model allows for understanding processes), and costs of collecting predictors data.

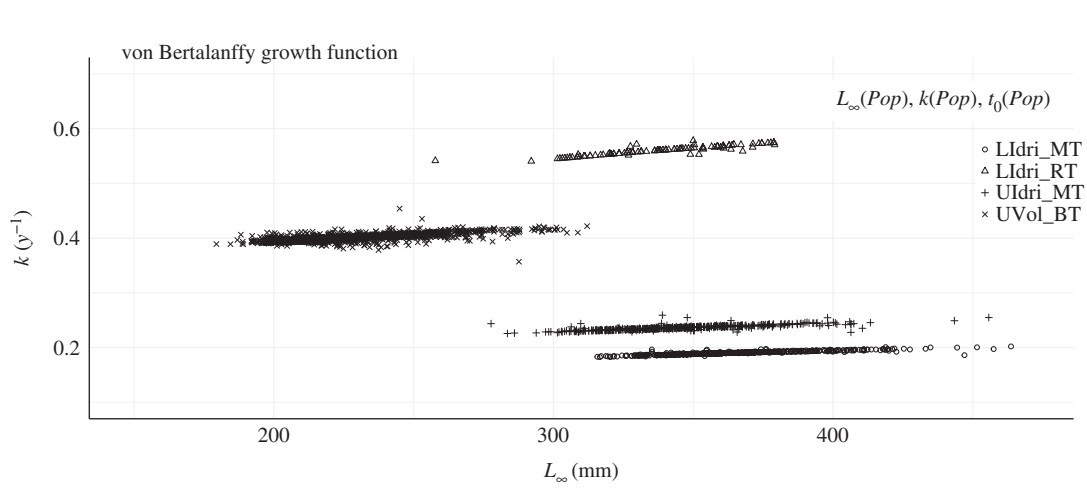

**Figure 4.** Correlation between $L_\infty$ and $k$ for the vBGF and $A$ and $k_G$ for the GGF for the best GGF (and equivalent von Bertalanffy growth function vBFG) model according to AIC (i.e. *Population* predicting all three parameters) when using the whole dataset. GGF (Pearson's *r, all p < 0.01*): LIdri_MT ($r = 0.65$); UIdri_MT (0.68); UVol_BT (0.63); LIdri_RT (0.33). vBGF: LIdri_MT ($r = 0.90$); UIdri_MT (0.83); UVol_BT (0.71); LIdri_RT (0.95).

The variation in growth and size that characterizes organisms can almost always be modelled retrospectively, but forecasting size-at-age is much more challenging. The limited number of attempts at predicting missing size observations or unobserved size-at-age and growth trajectories may also depend on the intrinsic unpredictability of the growth of some organisms. For instance, in species with environmental sexual determination and sexual dimorphism, such as eels [74], or when growth is faster later in life and is strongly determined by the environment (e.g. ocean growth of anadromous salmonids [75]), it may be impossible to accurately predict later portions of the growth trajectory when only observations early in life are available.

We have shown that for the four salmonid populations that we used as a model system, the best Gompertz and von Bertalanffy mixed-effects growth models allow one to use a single measurement early in the life of individual fish to obtain accurate predictions of their size-at-age in the future. In addition, the predictions made by the two models when using the same predictors for their parameters were basically the same; choosing between the best models of the two growth functions would have very few practical consequences for predictions of size-at-age of individuals. Some inaccurate predictions deserve further investigations, although they are to be expected when predicting the realizations of complex biological processes.

However, estimates of asymptotic size within populations were not correlated when estimated using the same Gompertz and von Bertalanffy growth models. For instance, for one highly ranked model in terms of AIC and predictive performance for both growth functions, the estimates of asymptotic size for rainbow trout provided by the Gompertz model were on average more than two times bigger than those provided by the von Bertalanffy model; in addition, for the same model, asymptotic size and growth rate at the individual level for rainbow trout were negatively correlated when using the

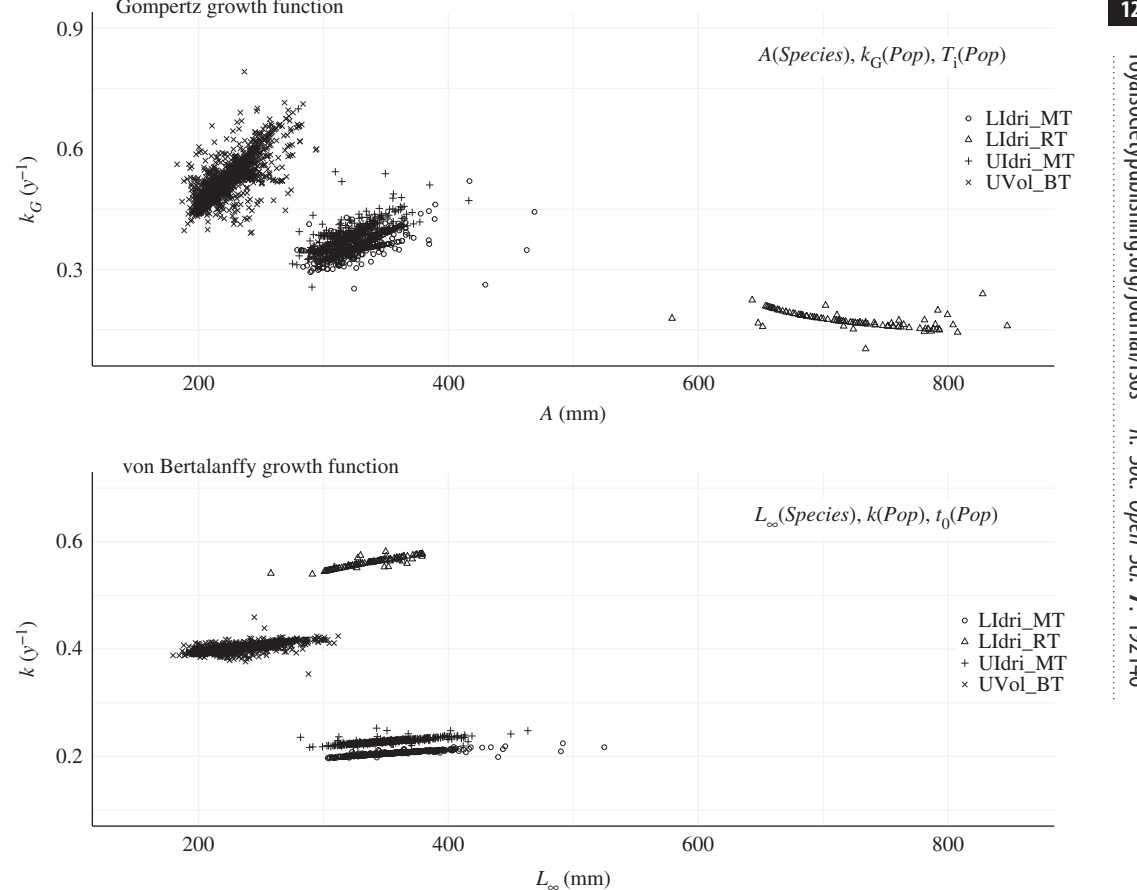

**Figure 5.** Correlation between $L_\infty$ and $k$ for the *vBGF* and $A$ and $k_G$ for the *GGF* for the best von Bertalanffy growth function (and equivalent Gompertz growth function) model according to prediction performance (i.e. *Species* predicting asymptotic size, and *Population* predicting the other two parameters). Parameters were estimated using the whole dataset. GGF (Pearson's *r, all p < 0.01*): LIdri_MT ($r = 0.65$); UIdri_MT (0.59); UVol_BT (0.76); LIdri_RT (−0.56). Von Bertalanffy growth function: LIdri_MT ($r = 0.91$); UIdri_MT (0.81); UVol_BT (0.73); LIdri_RT (0.91).

Gompertz model and positively correlated when using the von Bertalanffy model. Berkey [76] found that growth curve parameters estimated using the empirical Bayes method (although the same can be said of any estimation method for size-at-age models) are particularly sensitive to the endpoints of the growth trajectories. In our study, the population of rainbow trout had the smaller and sparser data among all four salmonid populations that we used in our system. However, wildly different combinations of parameters of these growth functions can result in some cases in very similar growth trajectories over a restricted time horizon [13]; this explains why the resulting predictions provided by the two models were basically the same, even when the inference on growth processes coming from the analysis of parameter estimates was different.

Although predictions from models of the two growth functions with similar performance on test datasets were highly correlated and barely distinguishable, the two growth functions typically provided different (in some cases, substantially different) estimates of asymptotic size. This suggests that distribution of size-at-age can be a more informative and stable-across-models measure and descriptor of growth than estimates of model parameters, especially when using or comparing different growth functions. In addition, it is often challenging to accurately estimate asymptotic size when only a few old individuals have been sampled. Other parametrizations of the growth equations [77] may also be explored.

Shohoji *et al.* [78] found that the classification of individuals into homogeneous groups was necessary to obtain accurate predictions of human lifetime growth. In our case, the classification either into population or species was sufficient to develop models that provided overall excellent predictions of the future growth of fish. When a single measurement early in life is sufficient to make accurate predictions of future growth, we can hypothesize that either the intrinsic growth potential of the individual, the environment experienced early in life, or a combination of the two largely determine

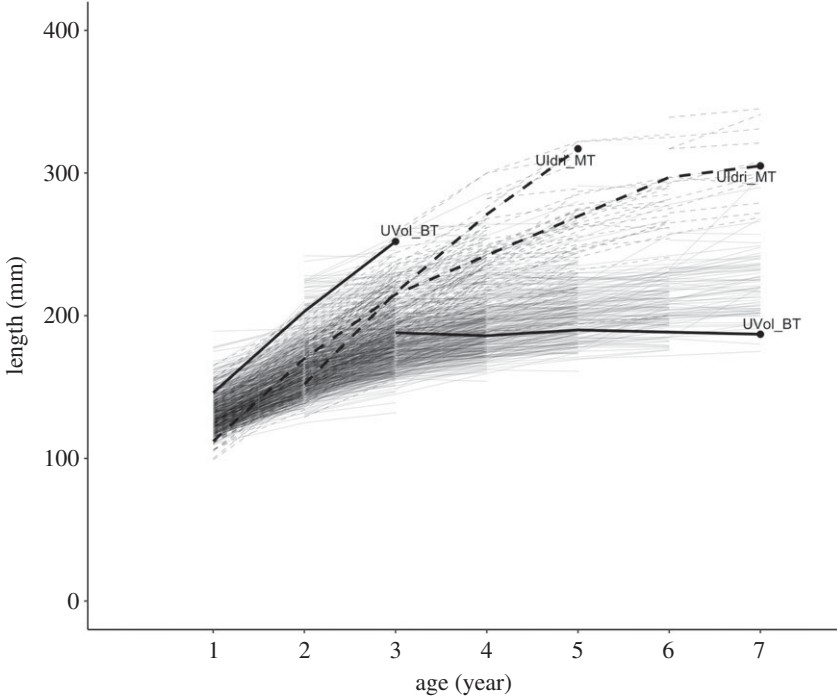

**Figure 6.** Growth trajectories of the four individuals (two from Uldri_MT and two from UVol_BT) that were most consistently the worst predicted (the black circle is the worst prediction for the individual) by both von Bertalanffy and Gompertz models. Light dashed lines are the empirical trajectories of all fish sampled in Uldri_MT and light solid lines are the empirical trajectories of all fish sampled in UVol_BT.

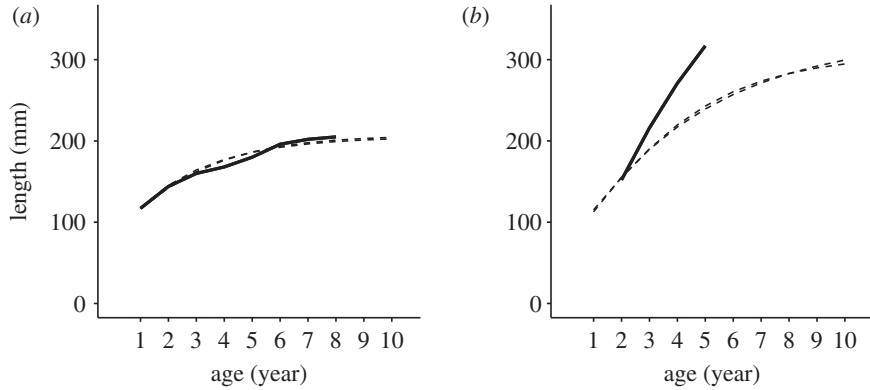

**Figure 7.** Example of good (*a*) and bad (*b*) predictions of growth (dashed lines) for GGF and vBGF models (all three parameters in either growth function with *Population* as predictor) for two individuals that have been sampled multiple times in UVol_BT (solid lines). The predictions of size-at-age provided by the GGF and vBGF were basically equivalent and based on size measured only at first sampling, at age 1+ for the individual in (*a*) and at age 2+ for the individual in (*b*).

the lifetime growth of individuals. Strong empirical evidence of early induced effects on later-in-life growth rate, life-history traits and behaviour of organisms is rapidly building up in the literature [79,80].

# 5. Conclusion

A better understanding of the evolutionary, physiological and life-history determinants of the differences in growth within and between individuals, populations, species and taxa will always be a major problem in biology. In the context of species that grow after sexual maturity and when used for predictive purposes, it appears that size-at-age mixed-effects growth models with time-invariant predictors sit in a favourable place on the surface that trades off accuracy, complexity, and biological interpretation of model parameters.

Ethics. All sampling work was approved by the Ministry of Agriculture, Forestry and Food of Republic of Slovenia and the Fisheries Research Institute of Slovenia. Original title of the Plan: RIBISKO - GOJITVENI NACRT za TOLMINSKI RIBISKI OKOLIS, razen Soce s pritoki od izvira do mosta v Cezsoco in Krnskega jezera, za obdobje 2006–2011. Sampling was supervised by the Tolmin Angling Association (Slovenia).

Data accessibility. Data and relevant code for this research work are stored in GitHub: https://github.com/ simonevincenzi/Heter/tree/master/raw_data [Data] and https://github.com/simonevincenzi/Growth_Models [Code], and have been archived within the Zenodo repository: https://doi.org/10.5281/zenodo.3697282.

Authors' contributions. S.V. conceived the ideas, designed the methods and ran the analyses; A.J.C. conceived and ran the marble trout project; A.J.C. and D.J. collected and curated the data; S.V. analysed the data; S.V. led the writing of the manuscript. All authors gave final approval for publication.

Competing interests. The authors declare no competing interests.

Funding. Financial support came from the MAVA Foundation.

Acknowledgements. We thank Marc Mangel for comments on the manuscript. We thank two anonymous reviewers for comments that greatly helped improve the paper.

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
