## [Reviewer comments · Royal Society Open Science]

Review History

RSOS-192146.R0 (Original submission)

Review form: Reviewer 1 (Mollie Brooks)

Is the manuscript scientifically sound in its present form?

Yes

Are the interpretations and conclusions justified by the results?

Yes

Is the language acceptable?

Yes

Do you have any ethical concerns with this paper?

No

Have you any concerns about statistical analyses in this paper?

No

Recommendation?

Accept with minor revision (please list in comments)

Comments to the Author(s)

This paper uses mark-recapture data from a long-term study of multiple species and multiple populations of fish in multiple locations. The sample sizes are impressive (n=209, 210, 17, and 1323 unique fish) as it is rare to find a long-term mark recapture study of fish. The results show that size hierarchies are likely to be maintained throughout ontogeny. Other studies (including some cited in the text) have shown that this has important consequences at the population level (also see Kraak et al. ICES Journal of Marine Science (2019), doi:10.1093/icesjms/fsz107). So the results will have broader reaching interest. Also, the growth models used in the paper are of broad interest.

General Comments

On the topic of the asymptotic size...in several places it seems that this parameter is not well observed and estimated (e.g. lines 465, 472, 483). You could cite (or in the future use the method from) Schnute and Fournier CJAS 1980 to make vBGF fitting less dependent on observing the asymptote. It's unfortunate that people continue to use the original ill-conditioned parameterization when a better alternative exists. A similar method may exist or could be developed for the GGF. Maybe a back-transformation exists to get the estimated asymptote so that the GGF and vBGF asymptotes could be compared. In any case, it should be acknowledged that the asymptote is not well estimated when not observed and better options should be explored in the future. Add CI lines to each point in Fig 2 in both directions (A and Linf) to show the uncertainty.

Maybe the cohort models would have better convergence if it was treated as a random effect, possibly (depending on biology) with separate levels for cohorts of different populations and species. This isn't absolutely necessary, but it's something to consider.

It's excellent that an attempt was made to make code and data available online. However, it isn't easy to reproduce results and see data. I tried to run the models, but it took too long (as helpfully documented in a README file). It would be nice if a simple example was presented where a single model was run and summarized. From what I could tell in my brief examination, all the R files run multiple models in an efficient, but difficult to read way; so it would be nice to have a simple example to start from. I did examine the TMB code and it looks reasonable and well commented.

Obtaining the organized data took some hunting. One way required running many lines of code in `gomp_parall_tmb.r`. It took me a while to find the last two lines of `Growth_Models/load_data.r` which read in the already organized data. I thought `load_data.R` would be irrelevant because the first few lines refer to "`~/Dropbox/Articoli/Limit_sampling`" so they caused errors for me. I tried to load data by double clicking objects in the data directory, but it appears that it requires `readRDS` statements like in `Growth_Models/load_data.r`. Why not save the organized data as Rdata instead of RDS? It would be easier to read into R. Or if you prefer to keep it as RDS, then make it more obvious how to load the organized data from `Growth_Models/data`.

Specific Comments

L17 Delete "determine".

L62-64 Is there a citation for this statement?

L65-68 The statement in parentheses is a full sentence, so it should probably be moved out of parentheses. Maybe line 188-192 could be moved up to the introduction near this statement since they are related.

L178 Give a citation for reading scales.

L236-240 Add semicolons instead of commas to increase level of division. "Where $L(t)$ is size at time t ; A (we might also call this parameter L_{∞} like in the von Bertalanffy growth function, since the two parameters both represent the asymptotic size); kG (in time⁻¹ units) is a coefficient of growth affecting the slope; and T_i is time at inflection, which in this formulation occurs when 37% of the final growth has been reached. "

L250 Next to "average" add "(mode)"

L265 Instead of the 3rd symbol, shouldn't it be phi as in line 267 (i.e. 3rd term on right hand side in each line of eqn 6)?

L267 Keep symbols in same order as above.

L321 Delete "results of".

L359-360. This seems more appropriate for the end of the methods section. Perhaps change "Results" to "Methods and results".

L389 Should be "terms".

L397 Should be "respect to".

L443 I'm not sure (and the reader might not understand) what this means.

L461-479 This should be one paragraph. At first it seemed that the thought ended on line 469. Do the CI show more uncertainty for this population? That would enhance the explanation.

L488 The meaning isn't clear and there may be a typo.

L501 grammatical mistake

Figure 1. White spaces between 4 panels could be smaller.

Figure 2. Add CI as stated above. If CI lines extend from the point in a + shape, then it may be necessary to replace the U_{ldri_MT} symbol + with another symbol. In caption, state what each point represents (12 models?) What 12 models? I thought there were 24 like in the table.

Figure 3. Spell out GGF and vBGF in titles as the shortened version isn't saving any space. I'm confused by the colors and line types. Can they be made more obvious? and/or explain them in the caption. Or even better, just directly label the 4 population means or modes in the graph.

To highlight the difference between Figs 4 and 5 (it's difficult to figure out), please add the model formula (in the same format as Table 1) directly to the Figures somewhere like the top-right corner. Also, spell out GGF and vBGF in titles as the shortened version isn't saving any space.

Review form: Reviewer 2

Is the manuscript scientifically sound in its present form?

Yes

Are the interpretations and conclusions justified by the results?

Yes

Is the language acceptable?

Yes

Do you have any ethical concerns with this paper?

No

Have you any concerns about statistical analyses in this paper?

No

Recommendation?

Accept with minor revision (please list in comments)

Comments to the Author(s)

Please see attached file (Appendix A).

Decision letter (RSOS-192146.R0)

10-Feb-2020

Dear Dr Vincenzi,

On behalf of the Editors, I am pleased to inform you that your Manuscript RSOS-192146 entitled "Biological and statistical interpretation of size-at-age, mixed-effects models of growth" has been accepted for publication in Royal Society Open Science subject to minor revision in accordance with the referee suggestions. Please find the referees' comments at the end of this email.

The reviewers and handling editors have recommended publication, but also suggest some minor revisions to your manuscript. Therefore, I invite you to respond to the comments and revise your manuscript.

- **Ethics statement**

- **Data accessibility**

<http://datadryad.org/submit?journalID=RSOS&manu=RSOS-192146>

- **Competing interests**

- **Authors' contributions**

- **Acknowledgements**

- **Funding statement**

Because the schedule for publication is very tight, it is a condition of publication that you submit the revised version of your manuscript before 19-Feb-2020. Please note that the revision deadline will expire at 00.00am on this date. If you do not think you will be able to meet this date please let me know immediately.

1) Identifying all the changes that have been made (for instance, in coloured highlight, in bold text, or tracked changes);

If your manuscript is newly submitted and subsequently accepted for publication, you will be asked to pay the article processing charge, unless you request a waiver and this is approved by Royal Society Publishing. You can find out more about the charges at <https://royalsocietypublishing.org/rsos/charges>. Should you have any queries, please contact openscience@royalsociety.org.

on behalf of Professor Len Thomas (Associate Editor) and Kevin Padian (Subject Editor)
openscience@royalsociety.org

Associate Editor Comments to Author (Professor Len Thomas):

Thank-you for submitting your manuscript to RSOS. I have received two thorough reviews back, and both reviewers are largely positive about your work, so I am happy to recommend acceptance with minor revision. In making your revision, please pay close attention to the reviewer comments, and give a point-by-point explanation of what changes you have made. In particular, regarding the comments of reviewer 1, please do consider implementing the Schnute and Fournier method; also please make your R code easier to run by providing a smaller example and using an .Rdata or other similar way to package it up (even an R package?). Regarding reviewer 2's comments, please consider including the additional contextual information they suggest. I look forward to seeing your revision and, should it prove satisfactory, don't anticipate the need to send it out for further peer review.

Reviewer comments to Author:

Reviewer: 1

Comments to the Author(s)

This paper uses mark-recapture data from a long-term study of multiple species and multiple populations of fish in multiple locations. The sample sizes are impressive ($n=209, 210, 17,$ and 1323 unique fish) as it is rare to find a long-term mark recapture study of fish. The results show that size hierarchies are likely to be maintained throughout ontogeny. Other studies (including some cited in the text) have shown that this has important consequences at the population level (also see Kraak et al. ICES Journal of Marine Science (2019), doi:10.1093/icesjms/fsz107). So the results will have broader reaching interest. Also, the growth models used in the paper are of broad interest.

General Comments

On the topic of the asymptotic size...in several places it seems that this parameter is not well observed and estimated (e.g. lines 465, 472, 483). You could cite (or in the future use the method from) Schnute and Fournier CJAS 1980 to make $vBGF$ fitting less dependent on observing the asymptote. It's unfortunate that people continue to use the original ill-conditioned parameterization when a better alternative exists. A similar method may exist or could be developed for the GGF. Maybe a back-transformation exists to get the estimated asymptote so that the GGF and $vBGF$ asymptotes could be compared. In any case, it should be acknowledged that the asymptote is not well estimated when not observed and better options should be explored in the future. Add CI lines to each point in Fig 2 in both directions (A and Linf) to show the uncertainty.

Maybe the cohort models would have better convergence if it was treated as a random effect, possibly (depending on biology) with separate levels for cohorts of different populations and species. This isn't absolutely necessary, but it's something to consider.

It's excellent that an attempt was made to make code and data available online. However, it isn't easy to reproduce results and see data. I tried to run the models, but it took too long (as helpfully documented in a ReadMe file). It would be nice if a simple example was presented where a single model was run and summarized. From what I could tell in my brief examination, all the R files run multiple models in an efficient, but difficult to read way; so it would be nice to have a simple example to start from. I did examine the TMB code and it looks reasonable and well commented.

Obtaining the organized data took some hunting. One way required running many lines of code in `gomp_parallel_tmb.r`. It took me a while to find the last two lines of

Growth_Models/load_data.r which read in the already organized data. I thought load_data.R would be irrelevant because the first few lines refer to "~/Dropbox/Articoli/Limit_sampling" so they caused errors for me. I tried to load data by double clicking objects in the data directory, but it appears that it requires readRDS statements like in Growth_Models/load_data.r. Why not save the organized data as Rdata instead of RDS? It would be easier to read into R. Or if you prefer to keep it as RDS, then make it more obvious how to load the organized data from Growth_Models/data.

Specific Comments

L17 Delete "determine".

L62-64 Is there a citation for this statement?

L65-68 The statement in parentheses is a full sentence, so it should probably be moved out of parentheses. Maybe line 188-192 could be moved up to the introduction near this statement since they are related.

L178 Give a citation for reading scales.

L236-240 Add semicolons instead of commas to increase level of division. "Where $L(t)$ is size at time t ; A (we might also call this parameter L_{∞} like in the von Bertalanffy growth function, since the two parameters both represent the asymptotic size); kG (in time-1 units) is a coefficient of growth affecting the slope; and T_i is time at inflection, which in this formulation occurs when 37% of the final growth has been reached. "

L250 Next to "average" add "(mode)"

L265 Instead of the 3rd symbol, shouldn't it be phi as in line 267 (i.e. 3rd term on right hand side in each line of eqn 6)?

L267 Keep symbols in same order as above.

L321 Delete "results of".

L359-360. This seems more appropriate for the end of the methods section. Perhaps change "Results" to "Methods and results".

L389 Should be "terms".

L397 Should be "respect to".

L443 I'm not sure (and the reader might not understand) what this means.

L461-479 This should be one paragraph. At first it seemed that the thought ended on line 469. Do the CI show more uncertainty for this population? That would enhance the explanation.

L488 The meaning isn't clear and there may be a typo.

L501 grammatical mistake

Figure 1. White spaces between 4 panels could be smaller.

Figure 2. Add CI as stated above. If CI lines extend from the point in a + shape, then it may be

necessary to replace the Uldri_MT symbol + with another symbol. In caption, state what each point represents (12 models?) What 12 models? I thought there were 24 like in the table.

Figure 3. Spell out GGF and vBGF in titles as the shortened version isn't saving any space. I'm confused by the colors and line types. Can they be made more obvious? and/or explain them in the caption. Or even better, just directly label the 4 population means or modes in the graph.

To highlight the difference between Figs 4 and 5 (it's difficult to figure out), please add the model formula (in the same format as Table 1) directly to the Figures somewhere like the top-right corner. Also, spell out GGF and vBGF in titles as the shortened version isn't saving any space.

Reviewer: 2

Comments to the Author(s)

Please see attached File.

Author's Response to Decision Letter for (RSOS-192146.R0)

See Appendix B.

Decision letter (RSOS-192146.R1)

16-Mar-2020

Dear Dr Vincenzi,

It is a pleasure to accept your manuscript entitled "Biological and statistical interpretation of size-at-age, mixed-effects models of growth" in its current form for publication in Royal Society Open Science. The comments of the reviewer(s) who reviewed your manuscript are included at the foot of this letter.

Kind regards,

Anita Kristiansen
Editorial Coordinator

on behalf of Professor Len Thomas (Associate Editor) and Kevin Padian (Subject Editor)
openscience@royalsociety.org

Associate Editor Comments to Author (Professor Len Thomas):

Comments to the Author:

Thank-you for providing a revised version. Although I am disappointed you have not included the Schnute and Fornier method I see you reference it, and have responded to all other points (I also can't find an issue on line 501). I am therefore happy to recommend acceptance of the paper. One small request is that you please acknowledge the two reviewers in your final version, in the acknowledgements section.

Appendix A

Manuscript ID RSOS-192146

Biological and statistical interpretation of size-at-age, mixed-effects models of growth

1 General Comments

This manuscript developed a mixed-effect model to evaluate the individual and group variations in body growth, with the application tagged freshwater trouts in Slovenian streams. In my opinion, this is an interesting and well written manuscript, with a detailed and useful modelling using highly valuable data. This manuscript shed light on the understanding of individual growth variability in indeterminate growers (does animals that never stop growing though there lifespan). In my opinion, I would recommend the publication this manuscript after satisfactory revision. My main comment is related with that key references on the subject in individual variability in fishes are missing. Because this paper is written in a general context, I would recommend to add and discuss the issue of individual growth variability in fishes in broader context with update references. Please see bellow for specific comments.

2 Specific comments

- The idea of modelling individual growth variability in fishes, started (as far as I know) with the pioneer work of Sainsbury (1980). I think some of the historical context of individual growth variability in fishes with benefit the manuscript. Wang and Thomas (1995) provides a good revision of the method (prior 1995) to assess growth viability in fishes. A more modern method is provided in Contreras-Reyes et al. (2008) using also, mixed effect model but in a Bayesian framework. I would recommend to start with those papers (and references therein) and give a broader context for your manuscript adding such information to the introduction/discussion.
- L55-58. I think is potentially misleading to refer to "growth model of vertebrates" as a classification to growth models. Note the vertebrate includes fishes (indeterminate growers) but also mammals (determinate

grower, in which growth cease after maturity). I would recommend to keep throughout the manuscript, the clarification of growth "determinate" or "indeterminate" as suggested in Charnov (1993). Gompertz and von Bertalanffy are models for indeterminate growers, as fish.

- L64 I Think this paragraph needs a reference.
- L69. I think VBGF, in length as use in the manuscript, is not a sigmoid function, because its does not have an inflection point as Gompertz has.
- L94-L97. A more efficient manner to obtain longitudinal data for modelling growth variability in fishes is via back-calculation of otolith radius, please see Contreras-Reyes et al. (2008).
- L193 (section 2.3.1). The vBGF with 3 parameters should be refereed as the "Specialized von Bertalanffy growth function".
- L252-L255. These lines are not entirely correct. Age-structured stock assessment methods are usually based on observations of catch-at-age with no direct effect of the growth parameters. When observation of catch at age are not available (just catch at length), then growth parameters are used directly on the stock assessment.
- L323. I think there is something wrong here. GFF and VBGF does not assume a particular structure of the data. Thus, defence of choosing just one month of sampling (September), should be based on rather more statistical background.

3 References

- Charnov EL. 1993. Life History Invariants. Oxford University Press, London.
- Contreras-Reyes J.E, F.O. López Quintero, R. Wiff. 2018. Bayesian modeling of individual growth variability using back-calculation: Application to pink cusk-eel (*Genypterus blacodes*) off Chile. Ecological Modelling. 385: 145-163.
- Sainsbury, K.J., 1980. Effect of individual variability on the von Bertalanffy growth equation. Can. J. Fish. Aquat. Sci. 37, 241-247.

- Wang, Y.G., Thomas, M. R., 1995. Accounting for individual variability in the von Bertalanffy growth model. *Can. J. Fish. Aquat. Sci.* 52, 1368-1375.

Appendix B

Associate Editor Comments to Author (Professor Len Thomas):

Thank-you for submitting your manuscript to RSOS. I have received two thorough reviews back, and both reviewers are largely positive about your work, so I am happy to recommend acceptance with minor revision. In making your revision, please pay close attention to the reviewer comments, and give a point-by-point explanation of what changes you have made. In particular, regarding the comments of reviewer 1, please do consider implementing the Schnute and Fournier method; also please make your R code easier to run by providing a smaller example and using an .Rdata or other similar way to package it up (even an R package?). Regarding reviewer 2's comments, please consider including the additional contextual information they suggest. I look forward to seeing your revision and, should it prove satisfactory, don't anticipate the need to send it out for further peer review.

- We appreciate the AE and Reviewer's comments, which greatly helped improve the second version of our manuscript.

Reviewer comments to Author:

Reviewer: 1

Comments to the Author(s)

This paper uses mark-recapture data from a long-term study of multiple species and multiple populations of fish in multiple locations. The sample sizes are impressive ($n=209, 210, 17,$ and 1323 unique fish) as it is rare to find a long-term mark recapture study of fish. The results show that size hierarchies are likely to be maintained throughout ontogeny. Other studies (including some cited in the text) have shown that this has important consequences at the population level (also see Kraak et al. ICES Journal of Marine Science (2019), doi:10.1093/icesjms/fsz107). So the results will have broader reaching interest. Also, the growth models used in the paper are of broad interest.

- We added Kraak et al. (2019) as one of the studies we cite with respect to including growth variation in management models.

General Comments

On the topic of the asymptotic size...in several places it seems that this parameter is not well observed and estimated (e.g. lines 465, 472, 483). You could cite (or in the future use the method from) Schnute and Fournier CJAS 1980 to make $vBGF$ fitting less dependent on observing the asymptote. It's unfortunate that people continue to use the original ill-conditioned parameterization when a better alternative exists. A similar

method may exist or could be developed for the GGF. Maybe a back-transformation exists to get the estimated asymptote so that the GGF and vBGF asymptotes could be compared. In any case, it should be acknowledged that the asymptote is not well estimated when not observed and better options should be explored in the future.

- We now added to the Discussion: "In addition, it is often challenging to accurately estimate asymptotic size when only a few old individuals have been sampled. Other parameterizations of the growth equations [77] should be explored."

Add CI lines to each point in Fig 2 in both directions (A and Linf) to show the uncertainty.

- Done.

Maybe the cohort models would have better convergence if it was treated as a random effect, possibly (depending on biology) with separate levels for cohorts of different populations and species. This isn't absolutely necessary, but it's something to consider.

- We have considered it and the consideration on the opportunity of including cohorts as random effects was discussed in previous papers that we wrote and we cite. Two main reasons for not including cohort as a random effect: 1) too few levels, 2) much longer times to convergence when including random effects other than individual random effects (and it is already taking quite a long time to fit these models). However, we provide the code (TMB) and researchers should be able to experiment with cohort random effects in their study systems, especially if they have more cohorts than we did.

It's excellent that an attempt was made to make code and data available online. However, it isn't easy to reproduce results and see data. I tried to run the models, but it took too long (as helpfully documented in a ReadMe file). It would be nice if a simple example was presented where a single model was run and summarized. From what I could tell in my brief examination, all the R files run multiple models in an efficient, but difficult to read way; so it would be nice to have a simple example to start from. I did examine the TMB code and it looks reasonable and well commented.

Obtaining the organized data took some hunting. One way required running many lines of code in `gomp_parall_tmb.r`. It took me a while to find the last two lines of `Growth_Models/load_data.r` which read in the already organized data. I thought `load_data.R` would be irrelevant because the first few lines refer to `"~/Dropbox/Articoli/Limit_sampling"` so they caused errors for me. I tried to load data by double clicking objects in the data directory, but it appears that it requires `readRDS` statements like in `Growth_Models/load_data.r`. Why not save the organized data as Rdata instead of RDS? It would be easier to read into R. Or if you prefer to keep it as

RDS, then make it more obvious how to load the organized data from Growth_Models/data.

- Following Reviewer's feedback, we have added a self-contained script for a few models (20 mins to fit on a modern MacBook). We have also better described the organization of the github repo and we will continue working on it. We prefer not to use a Rdata object because it would be too big and might create problems during loading. The .RDS type is a standard way of saving R objects, but the automatic creation of a R object by clicking on the .RDS happens only in RStudio as far as we know, otherwise the command readRDS needs to be used (as described in the repo).

Specific Comments

L17 Delete "determine".

- Done.

L62-64 Is there a citation for this statement?

- Yes, we added as reference: "Vincenzi S, Crivelli AJ, Munch S, Skaug HJ, Mangel M. 2016 Trade-offs between accuracy and interpretability in von Bertalanffy random-effects models of growth. *Ecol. Appl.* **26**, 1535–1552. (doi:10.1890/15-1177)".

L65-68 The statement in parentheses is a full sentence, so it should probably be moved out of parentheses. Maybe line 188-192 could be moved up to the introduction near this statement since they are related.

- We moved the statement out of parentheses.

L178 Give a citation for reading scales.

- Added.

L236-240 Add semicolons instead of commas to increase level of division. "Where $L(t)$ is size at time t ; A (we might also call this parameter L_{∞} like in the von Bertalanffy growth function, since the two parameters both represent the asymptotic size); kG (in time^{-1} units) is a coefficient of growth affecting the slope; and T_i is time at inflection, which in this formulation occurs when 37% of the final growth has been reached. "

- Added semi-colons.

L250 Next to "average" add "(mode)"

- Done.

L265 Instead of the 3rd symbol, shouldn't it be phi as in line 267 (i.e. 3rd term on right hand side in each line of eqn 6)?

- It should be phi. We have now changed to phi.

L267 Keep symbols in same order as above.

- Done.

L321 Delete "results of".

- Done.

L359-360. This seems more appropriate for the end of the methods section. Perhaps change "Results" to "Methods and results".

- We moved the sentence to the end of Methods.

L389 Should be "terms".

- Done.

L397 Should be "respect to".

- Done.

L443 I'm not sure (and the reader might not understand) what this means.

- We deleted "maintaining data pipelines", which is a term used more in industry than in academia and could generate confusion.

L461-479 This should be one paragraph. At first it seemed that the thought ended on line 469. Do the CI show more uncertainty for this population? That would enhance the explanation.

- Done, the formatting was wrong. It is now a single paragraph.

L488 The meaning isn't clear and there may be a typo.

- We deleted the part within parentheses, it was not clear.

L501 grammatical mistake

- Unfortunately, we didn't see the grammatical mistake.

Figure 1. White spaces between 4 panels could be smaller.

- We have now reduced the white space between panels.

Figure 2. Add CI as stated above. If CI lines extend from the point in a + shape, then it may be necessary to replace the Uldri_MT symbol + with another symbol. In caption, state what each point represents (12 models?) What 12 models? I thought there were 24 like in the table.

- We now added the 95% CI for L_{∞} and A . In Table 1, there are 12 GGF and 12 vBGF growth models. To clarify, we write: "The mean [min, max] of the ratio L_{∞} / A across the estimates for the 12 von Bertalanffy and 12 Gompertz growth models were..."

Figure 3. Spell out GGF and vBGF in titles as the shortened version isn't saving any space. I'm confused by the colors and line types. Can they be made more obvious? and/or explain them in the caption. Or even better, just directly label the 4 population means or modes in the graph.

- Done, we spelled out GGF and vBGF and directly labeled the density curves.

To highlight the difference between Figs 4 and 5 (it's difficult to figure out), please add the model formula (in the same format as Table 1) directly to the Figures somewhere like the top-right corner. Also, spell out GGF and vBGF in titles as the shortened version isn't saving any space.

- Done, we spelled out GGF and vBGF and we added the model formula.

REVIEWER #2

Manuscript ID RSOS-192146

Biological and statistical interpretation of size-at-age, mixed-effects models of growth

1 General Comments

This manuscript developed a mixed-effect model to evaluate the individual and group variations in body growth, with the application tagged freshwater trouts in Slovenian streams. In my opinion, this is an interesting and well written manuscript, with a detailed and useful modelling using highly valuable data. This manuscript shed light on the understanding of individual growth variability in indeterminate growers (does animals that never stop growing though there lifespan). In my opinion, I would recommend the publication this manuscript after satisfactory revision. My main comment is related with that key references on the subject in individual variability in fishes are missing. Because this paper is written in a general context, I would recommend to add and discuss the issue of individual growth variability in fishes in broader context with update references. Please see below for specific comments.

2 Specific comments

- The idea of modelling individual growth variability in fishes, started (as far as I know) with the pioneer work of Sainsbury (1980). I think some of the historical context of individual growth variability in fishes with benefit the manuscript. Wang and Thomas (1995) provides a good revision of the method (prior 1995) to assess growth viability in fishes. A more modern method is provided in Contreras-Reyes et al. (2008) using also, mixed effect model but in a Bayesian framework. I would recommend to start with those papers (and references therein) and give a broader context for your manuscript adding such information to the introduction/discussion.

- We have added context at the end of the Introduction. We now write: “ Since [23] developed von Bertalanffy equations for individual growth, several growth models describing or accounting for individual and group variation in growth, along with algorithms for their fitting, have been developed [24]. Recently, [25] developed a Bayesian state-space framework, there applied to the von Bertalanffy growth function, for modeling growth that allows for intrinsic individual variation in traits, a shared environment, process stochasticity, and measurement error. [26] developed a framework that comprised modeling of individual variability of size-at-age, Bayesian inference, and distribution of errors from the Student- t model.”

- L55-58. I think is potentially misleading to refer to ”growth model of vertebrates” as a classification to growth models. Note the vertebrate includes fishes (indeterminate growers) but also mammals (determinate grower, in which growth cease after maturity). I would recommend to keep throughout the manuscript, the clarification of growth ”determinate” or ”indeterminate” as suggested in Charnov (1993). Gompertz and von Bertalanffy are models for indeterminate growers, as fish.

- We now write: “Structured or parametric models for growth of organisms that continue to increase in size after sexual maturity (i.e., indeterminate growers, [6]) imply a basic functional form: size increases monotonically with time and it usually tends to an upper asymptote later on in life.”
- L64 I Think this paragraph needs a reference.
 - Reference added.
- L69. I think VBGF, in length as use in the manuscript, is not a sigmoid function, because its does not have an inflection point as Gompertz has.
 - It is normally considered a sigmoid, but we have now deleted that reference to sigmoid curves since it can generate confusion.
- L94-L97. A more efficient manner to obtain longitudinal data for modelling growth variability in fishes is via back-calculation of otolith radius, please see Contreras-Reyes et al. (2008).
 - Yes, it is possible to use otolith too, although there is greater uncertainty on the size measures and the individual has to be killed, which cannot be done for marble trout living in Slovenian streams. Tagging in the wild is much more time consuming, but when done properly, measures tend to be quite accurate.
- L193 (section 2.3.1). The vBGF with 3 parameters should be refereed as the ”Specialized von Bertalanffy growth function”.
 - We now write:” The forms of the vBGF in Eqs. (3) and (4), which are derived following the assumption of anabolism scaling with mass to the $2/3$ power and mass scaling with the cube of size, are referred to as the “specialized” vBGF [45]. For simplicity, in this paper we will simply use the term vBGF for the “specialized” vBGF.“
- L252-L255. These lines are not entirely correct. Age-structured stock assessment methods are usually based on observations of catch-at-age with no direct effect of the growth parameters. When observation of catch at age are not available (just catch at length), then growth parameters are used directly on the stock assessment.
 - We now write: “Age-structured stock assessment methods **can also** be based on sizes-at-age estimations, which are often derived from parameters of the von Bertalanffy growth model for that species [44]”.
- L323. I think there is something wrong here. GFF and VBGF does not assume a particular structure of the data. Thus, defence of choosing

just one month of sampling (September), should be based on rather more statistical background.

- Yes, there is no explicit assumption on the structure of the data. We now write more precisely: “We found empirically that for both the GFF and vBGF, the fitting algorithm converges more reliably when size data are evenly spaced in time”.

3 References

- Charnov EL. 1993. Life History Invariants. Oxford University Press, London.
 - Contreras-Reyes J.E, F.O. L´opez Quintero, R. Wiff. 2018. Bayesian modeling of individual growth variability using back-calculation: Application to pink cusk-eel (*Genypterus blacodes*) off Chile. *Ecological Modelling*. 385: 145-163.
 - Sainsbury, K.J., 1980. Effect of individual variability on the von Bertalanffy growth equation. *Can. J. Fish. Aquat. Sci.* 37, 241-247.
- 2
- Wang, Y.G., Thomas, M. R., 1995. Accounting for individual variability in the von Bertalanffy growth model. *Can. J. Fish. Aquat. Sci.* 52, 1368-1375